# Advances of Research on Dual-Frequency Solid-State Lasers for Synthetic-Wave Absolute-Distance Interferometry

**DOI:** 10.3390/s23063206

**Published:** 2023-03-17

**Authors:** Mingxing Jiao, Fei Jiang, Junhong Xing, Yun Liu, Tianhong Lian, Jianning Liu, Guangtao Li

**Affiliations:** School of Mechanical and Precision Instrumental Engineering, Xi’an University of Technology, Xi’an 710048, China

**Keywords:** dual-frequency solid-state laser, frequency-difference tuning, frequency-difference stabilization, quadrature-demodulated Pound–Drever–Hall method, synthetic-wave absolute-distance interferometry

## Abstract

Frequency-difference-stabilized dual-frequency solid-state lasers with tunable and large frequency difference have become an ideal light source for the high-accuracy absolute-distance interferometric system due to their stable multistage synthetic wavelengths. In this work, the advances in research on oscillation principles and key technologies of the different kinds of dual-frequency solid-state lasers are reviewed, including birefringent dual-frequency solid-state lasers, biaxial and two-cavity dual-frequency solid-state lasers. The system composition, operating principle, and some main experimental results are briefly introduced. Several typical frequency-difference stabilizing systems for dual-frequency solid-state lasers are introduced and analyzed. The main development trends of research on dual-frequency solid-state lasers are predicted.

## 1. Introduction

The synthetic-wave absolute-distance interferometric measurement is a kind of high-accuracy no-guideway ranging technology based on the fraction-coincident method of the interference fringes, which provides an effective way to solve the technical problems of precision measurement and ultra-precision measurement of large-dimension workpieces. At present, the dual-frequency lasers [1,2] and optical-frequency combs [3,4,5,6] are normally used as the light sources for synthetic-wave absolute-distance interferometric systems, in which dual-frequency lasers have attracted great attention and strong research interests of scholars worldwide due to the advantages of simple structure and low cost. As the typical representatives of dual-frequency lasers, the Zeeman dual-frequency He-Ne laser at 632.8 nm [7], the two-longitudinal-mode He-Ne laser at 632.8 nm [8], and the birefringent dual-frequency He-Ne laser at 632.8 nm [9] have been successfully applied to the synthetic-wave absolute-distance interferometric system. However, the frequency differences of these dual-frequency lasers are generally <1 GHz due to the fact that the fluorescence linewidth of Ne atoms is relatively narrow (about 1500 MHz), and the corresponding synthetic wavelength is >300 mm; accordingly, it is difficult to improve the accuracy of the absolute-distance measurement. It is found that the fluorescence linewidth of solid-state crystals is much wider than that of the He-Ne gas medium, and a dual-frequency solid-state laser with a frequency difference larger than several tens of GHz or up to several THz can form much smaller synthetic wavelengths, so that the accuracy of the absolute-distance measurement can be significantly improved. In addition, the integer order of the synthetic-wave-interference fringes can be uniquely determined, provided that the preliminary measurement error of the measured distance is less than one fourth of the synthetic wavelength; additionally, when the frequency difference of the dual-frequency laser decreases, the corresponding synthetic wavelength becomes longer, and, accordingly, the preliminary measurement of the measured distance can be performed easily. Evidently different multistage synthetic wavelengths can be generated using a dual-frequency solid-state laser with tunable and large frequency difference as a light source. A step-by-step refined measurement method can be used to realize a high-precision absolute-distance measurement, and the final accuracy of the absolute-distance measurement mainly depends on the magnitude of the minimum synthetic wavelength and its stability, that is, it is determined by the maximum frequency difference of the dual-frequency solid-state laser source and its stability. In order to achieve the absolute-distance measurement accuracy to an order greater than 10^−6^, the frequency-difference stability of the dual-frequency solid-state laser must be an order of 10^−7^ or better. Therefore, the frequency-difference-stabilized dual-frequency solid-state laser with tunable and large frequency difference can be used as an ideal light source for the synthetic-wave absolute-distance interferometric system [10,11,12,13].

Many experts and scholars worldwide have investigated the oscillating principles and key technologies of dual-frequency solid-state lasers with tunable and large frequency difference since 2000 [14,15,16,17,18,19], and they have successfully developed a series of new dual-frequency solid-state lasers, such as the birefringent dual-frequency solid-state laser [20,21,22,23,24,25,26,27,28,29,30], the biaxial dual-frequency solid-state laser [31,32], the two-cavity dual-frequency Nd:YAG laser (TCDFL) [33,34,35,36,37,38,39,40], and so on. Especially in the recent years, our research group has made important progresses in the research and development of the TCDFL and its quadrature-demodulated Pound–Drever–Hall (QD-PDH) frequency-stabilizing technology [41,42], and the QD-PDH frequency-stabilizing method has been applied to the frequency-difference stabilization of the TCDFL with a frequency difference of 24 GHz at 1064 nm [43,44]. This frequency-difference-stabilized TCDFL at 1064 nm has been successfully used as the light source for a synthetic-wave absolute-distance interferometric system [45].

## 2. Simultaneous Oscillation and Frequency-Difference Tuning of Orthogonally and Linearly Polarized Dual-Frequency Solid-State Laser

It is known that a dual-frequency laser output can be obtained by inserting a longitudinal-mode-splitting element into the laser cavity for the single longitudinal mode to be split into two orthogonally polarized components, provided that the laser meets the requirement of single-longitudinal-mode operation. At present, the commonly used methods of the laser single-longitudinal-mode selection for solid-state lasers mainly include the birefringent filter method, Fabry–Perot (F-P) etalon method, short-cavity method, twisted-mode-cavity method, ring traveling-wave-cavity method, etc. Additionally, the laser longitudinal-mode-splitting methods for solid-state lasers mainly include the birefringence splitting method and polarization splitting method. Due to the significant advantages of dual-frequency solid-state lasers, several different schemes of dual-frequency solid-state lasers have been designed and experimentally investigated by experts worldwide, which are based on the principles of single-longitudinal-mode selection by birefringent filter or intracavity F-P etalon, and the orthogonally and linearly polarized dual-frequency laser with tunable frequency difference has been simultaneously oscillated and output.

### 2.1. Birefringent Dual-Frequency Solid-State Lasers

#### 2.1.1. Natural Birefringent Dual-Frequency Solid-State Lasers

Dual-frequency solid-state microchip lasers have the advantages of large frequency difference, high pumping efficiency, high beam quality, and narrow laser linewidth. To obtain a dual-frequency laser output with a tunable and large frequency difference, in 2009, the research group of Profs. A. Mckay and J. M. Dawes of Macquarie University reported a diode-pumped dual-frequency microchip Nd:YAG ceramic laser at 1064 nm [20], as shown in Figure 1. This laser consisted of a 0.25 mm-long highly doped (4%) ceramic Nd:YAG plate and two zeroth-order quarter-wave plates, which were optically bonded to BK7 glass substrates with planar 1064 nm resonator mirror coatings on each waveplate. The ceramic plate was glued to the input-waveplate mirror, and the output-waveplate mirror was positioned close to the ceramic. The overall cavity length was estimated to be 300 μm, so the laser at 1064 nm was forced to oscillate in single-longitudinal mode. Due to the natural birefringence effect, the single longitudinal mode was split and the orthogonally and linearly polarized dual-frequency laser at 1064 nm was oscillated and output. Figure 2 shows the experimentally observed typical spectrums of the two simultaneously lasing narrow-linewidth-longitudinal modes with approximately equal optical power, in which Figure 2a–c are the oscillating modes of the dual-frequency laser with different frequency difference, respectively. At a frequency difference > 120 GHz [see Figure 2c], the optical power in both polarizations decreased due to the limited gain near the edges of the spectral-gain linewidth, which ultimately limited the tuning range of the frequency difference. By tuning the relative angles between the principal axes of the two quarter-wave plates, the frequency difference was tuned linearly from a few gigahertz to over 150 GHz, as shown in Figure 3.

In 2007, the research group of Prof. Zhao Changming of the Beijing Institute of Technology reported a diode-pumped tunable dual-frequency Nd:YAG laser at 1064 nm [21], as shown in Figure 4, in which an intracavity-fused quartz-made F-P etalon with a thickness of 0.5 mm was used as the laser longitudinal-mode selector, and two zeroth-order quarter-wave plates were vertically inserted into the laser cavity. As a result, an orthogonally polarized dual-frequency laser at 1064 nm was obtained, and the frequency difference of the dual-frequency laser was tuned by adjusting the relative angles between the principal axes of the two quarter-wave plates. The experimentally observed oscillating mode spectrum of the dual-frequency laser at 1064 nm was obtained by the use of a confocal scanning F-P interferometer, as shown in Figure 5, in which it can be seen that both longitudinal modes oscillated simultaneously in a single-longitudinal-mode, and the frequency difference of the dual-frequency laser at 1064 nm was tuned in a range from 50 MHz to 1.3 GHz. In 2010, the group also reported a diode-pumped coupled-cavity dual-frequency Nd:YAG laser at 1064 nm with tunable frequency difference [22], as shown in Figure 6. The single-longitudinal-mode oscillation was realized from a coupled-cavity setup consisting of the input-cavity mirror and a fused quartz-fabricated F-P etalon with a thickness of 1 mm and an effective reflectivity of 4%. The two quarter-wave plates were inserted into a cavity so as to split the laser-longitudinal mode, and the frequency difference was tuned in a range from 0 to 1.1 GHz by changing the angle between the principal axes of the two quarter-wave plates.

In 2001, the research group of Prof. Zhang Shulian of the Tsinghua University reported a diode-pumped birefringent dual-frequency Nd:YAG laser at 1064 nm [23], as shown in Figure 7, in which the resonant cavity of the laser contained a piece of crystal-quartz-made birefringent F-P etalon (BFPE), serving as both a selector and a splitter of laser longitudinal modes. Because of the intracavity birefringent effect, each longitudinal mode was split into two linearly and orthogonally polarized components, i.e., ordinary mode (o-mode) and extraordinary mode (e-mode), and the unique transmission maximum of the etalon within the overall lasing bandwidth of the Nd:YAG laser was also split into two transmission maxima, namely ordinary peak (o-peak) and extraordinary peak (e-peak). The simultaneous operation of the two longitudinal modes could be obtained by making an o-mode and an e-mode coincide with the central positions of the o-peak and e-peak, respectively. A piece of BFPE with a geometrical thickness of 645 mm and a cut-angle of 10° was designed and fabricated, which was placed in a 40 mm-long cavity of the diode-pumped Nd:YAG laser. As a result, the orthogonally and linearly polarized dual-frequency laser at 1064 nm was output by slightly adjusting the tilt angle of the BFPE element, and a frequency difference of approximately 2 GHz was observed experimentally by a confocal F-P-scanning interferometer with a free spectral range of 4 GHz.

In 2018, the research group of Prof. Jiao Mingxing of the Xi’an University of Technology designed and reported a diode-pumped dual-frequency Nd:YAG laser with a detuning twisted-mode cavity at 1064 nm [24], as shown in Figure 8, in which the polarizing beam splitter (PBS) was used as the polarizer and two quarter-wave plates were placed on each end of the Nd:YAG crystal. The laser oscillated in linearly polarized single-longitudinal-mode when the principal axes of the two quarter-wave plates were perpendicular to each other, and the single-longitudinal-mode was split into two orthogonally and linearly polarized modes when the twisted-mode cavity was detuned. As a result, an orthogonally polarized dual-frequency laser at 1064 nm was obtained, and the frequency difference was continuously tuned over the whole cavity free spectral range by rotating one of the quarter-wave plates in the plane perpendicular to the cavity axis. The oscillating-mode spectrum of the dual-frequency laser was observed experimentally by the use of a confocal scanning F-P interferometer, as shown in Figure 9. The dependence of the mode-splitting magnitude on the rotation angle of the quarter-wave plate was obtained experimentally, as indicated in Figure 10, and the largest frequency difference was about 3 GHz, determined by the laser cavity length.

#### 2.1.2. Electro-Optical and Thermo-Optical Birefringent Dual-Frequency Solid-State Lasers

Compared to the crystal-quartz-made BFPE described above, an electro-optical birefringent F-P etalon (EO-BFPE) can also be used as not only the single-longitudinal-mode selector but also as the splitter of the laser-longitudinal mode. On this foundation, in 2007, the research group of Julien Le Gouet of Thales Research & Technology, France, reported an electro-optical birefringent dual-frequency Nd:YAG laser [25], as shown in Figure 11, in which a 400 μm-thick EO-BFPE made from lead zirconate tantalate ceramic was included in the laser cavity with an optical length of 14 mm. As a result, an orthogonally and linearly polarized dual-frequency laser at 1064 nm was obtained, and the frequency difference was discontinuously tuned in a range from 10 to 127 GHz when the direct voltage applied to the EO-BFPE was changed from 0 to 499 V. The typical oscillating spectrum of the dual-frequency laser with different frequency difference was observed experimentally, as shown in Figure 12.

The research group of Prof. M. Brunel of the Université de Rennes 1 has been devoted to dual-frequency solid-state laser technologies for a long time (since 1997). Using the thermo-optical birefringent effect of the intracavity LiTaO_3_ crystal, in 2005, the group reported a microchip dual-frequency laser containing an erbium–ytterbium glass medium with a thickness of 190 μm and a LiTaO_3_ crystal with a thickness of 130 μm [26], as shown in Figure 13. The cavity length was less than 0.5 mm; thus, the orthogonally and linearly polarized dual-frequency laser at 1530 nm was simultaneously oscillated and output, and the frequency difference of the dual-frequency laser was continuously tuned from a few GHz to more than 50 GHz when the temperature of the LiTaO_3_ crystal changed from nearly 45 to 10 °C, as shown in Figure 14.

In 2008, the group also reported a diode-pumped dual-frequency solid-state laser [27], as shown in Figure 15, in which an erbium–ytterbium glass with a thickness of 0.75 mm was used as the gain medium. This laser also contained a 250 μm-thick F-P etalon acting as the single-longitudinal-mode selector and a 1 mm-thick LiTaO_3_ crystal serving as the longitudinal-mode splitter. As a result, an orthogonally polarized dual-frequency laser at 1530 nm was obtained, and the frequency difference of the dual-frequency laser was tuned in a range from 100 MHz to 20 GHz when the temperature of the LiTaO_3_ crystal varied from 22 to 42 °C, as shown in Figure 16.

#### 2.1.3. Stress Birefringent Dual-Frequency Solid-State Lasers

In 2003, the research group of Prof. V. G. Gudelev of the National Academy of Sciences of Belarus reported a diode-pumped tunable dual-frequency Nd:YAG laser at 1064 nm with coupled resonators [28], as shown in Figure 17. The coupled resonators consisted of the two sides of the Nd:YAG crystal and the spherical output coupler of OC, and the fundamental cavity formed by the left side of the Nd:YAG crystal and the OC was forced to oscillate in either single-longitudinal mode or two longitudinal modes, depending on the geometrical cavity length of L. On the one hand, in the circumstance of single-longitudinal-mode oscillation, the laser mode was split due to the intracavity stress birefringence induced by the applied force of F to the OC element; therefore, an orthogonally and linearly polarized dual-frequency laser at 1064 nm was obtained, and the frequency difference was continuously tuned in a range from 50 MHz to 2.4 GHz when changing the applied force. The oscillating-mode spectrum of the dual-frequency laser with different frequency difference was observed experimentally, as shown in Figure 18. On the other hand, in the circumstance of two-longitudinal-mode oscillation of the fundamental cavity, each mode was split into two orthogonally and linearly polarized modes when the applied force existed, the mode-splitting magnitude increased with the applied force. When the mode splitting magnitude reached to a certain value, the system output a beam of orthogonally and linearly polarized dual-frequency laser at 1064 nm, the maximum frequency difference of which being 8.4 GHz, as shown in Figure 19.

In 2019, the research group of Prof. Zhang Shulian of the Tsinghua University reported a diode-pumped dual-frequency Nd:YAG microchip laser at 1064 nm [29], as shown in Figure 20, in which a <111>-cut Nd:YAG microchip with 1 ± 0.1% neodymium doping in a circular shape was included. The diameter of the crystal was 5 ± 0.05 mm and the thickness of the microchip was 1 ± 0.1 mm to reduce the multiple longitudinal modes inside the laser output. The two surfaces were parallel and coated with high-reflectivity dielectric films acting as the laser-cavity mirrors, so the laser at 1064 nm was forced to oscillate in single-longitudinal mode. Based on the effect of stress birefringence, the single-oscillated-laser mode was split into two orthogonally polarized components; thus, the orthogonally and linearly polarized dual-frequency laser output at 1064 nm was obtained, as shown in Figure 21. It can be seen that the frequency difference was in the scale of tens of MHz, determined by the internal stress inside the microchip laser. In 2021, the group also studied a microchip Nd:YAG dual-frequency laser with a frequency difference of 17.4 MHz [30], as shown in Figure 22, which was formed by the stress-induced birefringence in the microchip itself. The <111>-cut, 1%-doped and quasi-isotropic Nd:YAG crystal was processed into a plate with a diameter of 2.8 mm and a thickness of 1 mm. Both faces of the chip were dielectric-coated to form a monolithic-resonant cavity; only one longitudinal mode resonated in the cavity, which was split into two monofrequency components due to the stress-induced birefringence in the Nd:YAG crystal. Thus, the laser emitted an orthogonally polarized dual-frequency laser with an output power of 4.7 mW at 1064 nm. The polarization directions of these two components coincided with the principal stress directions, respectively, and the frequency difference between them was proportional to the difference between the magnitudes of two principal stresses.

### 2.2. Biaxial Dual-Frequency Solid-State Lasers

In order to obtain a dual-frequency laser output with a tunable and large frequency difference, in 2004, the research group of R. Czarny of Thales Research & Technology, France, reported a diode-pumped dual-frequency KGd(WO_4_)_2_ Yb^3+^-doped laser for CW-THz generation [32], as shown in Figure 23, which included a 2 mm-long Yb:KGW active medium doped with 5% Yb. The 60 mm-long laser cavity was formed by two mirrors of M_1_ and M_2_. Spatial separation between the two eigenmodes inside a portion of the cavity was obtained using an AR-coated 10-mm long YVO_4_ crystal cut at 45° of its optical axes. It led to two cross-polarized eigenmodes (o-mode and e-mode). The Yb:KGW crystal was oriented to maximize gain of eigenstate (e), and a quartz-made half waveplate λ/2 was inserted in the o-mode path. To independently tune the optical frequencies associated to each polarization eigenstate, two noncoated 150 μm-thick glass etalons of E_o_ and E_e_ were placed in the o-mode path and e-mode path, respectively. The single-longitudinal-mode operation of both eigenstates was obtained thanks to a quite short cavity (about 60 mm) and an additional 1 mm-thick glass etalon E_c_ inserted in the common path; thus, the orthogonally and linearly polarized dual-frequency laser at 1030 nm was obtained, as shown in Figure 24. The frequency difference between the two modes was step-tunable from DC to 3.1 THz. A maximum total optical output power of 120 mW was obtained with a beat-note linewidth narrower than 30 kHz.

### 2.3. Two-Cavity Dual-Frequency Solid-State Lasers

According to the principle of single-longitudinal-mode selection by the use of a birefringent filter, the output of a high-power single-frequency laser is difficult to be obtained, due to the fact that the mode-selecting ability of the traditional birefringent filter is too low using a Brewster plate (BP) as a polarizer. Although the mode-selecting ability of the birefringent filter can be effectively enhanced by increasing the number of BP, some problems inevitably occur, such as complex laser structure, difficult adjustment of the laser cavity, etc. To overcome the shortcomings of the limited mode-selecting ability of the traditional birefringent filter, in 2006, Prof. Jiao Mingxing proposed a new birefringent filter in which a PBS was employed as the polarizer, and a TCDFL with a large frequency difference was invented and investigated [33,34]. Since then, our group have designed and researched several TCDFLs using different principles of single-longitudinal-mode selection, such as birefringent filter, F-P etalon, twisted-mode cavity, etc.

#### 2.3.1. TCDFLs Using Birefringent Filter

In order to obtain a tunable dual-frequency laser with a large frequency difference, our group designed and investigated a diode-pumped dual-frequency Nd:YAG laser with a tunable frequency difference [35], as shown in Figure 25, using the light-splitting and polarizing functions of the PBS to form perpendicular linear and right-angle standing-wave cavities. Both cavities included a birefringent output coupler (BOC) fabricated from calcite crystal, and they employed a birefringent filter of PBS-BOC acting as laser-longitudinal-mode selectors. The p and s components of the 1064 nm laser light oscillating in single-longitudinal-mode were observed experimentally, as shown in Figure 26. The frequency difference of the dual-frequency laser was tuned by adjusting the tilt angles of the BOC, and the oscillating spectrums of the dual-frequency laser at 1064 nm were observed, as shown in Figure 27. It can be seen that the frequency difference was tunable in a range from 11.9 GHz to 148.4 GHz, and the maximum frequency difference reached the fluorescence line a width of Nd:YAG.

Moreover, a diode-pumped TCDFL with a tunable and large frequency difference at 1064 nm was designed and investigated [36], as shown in Figure 28. Both cavities included a birefringent filter of PBS-BC, in which the birefringent crystal (BC) was a piece of a half-wave plate with a thickness of 0.7 mm. The p-polarized and s-polarized components of the laser at 1064 nm were forced to oscillate simultaneously in single-longitudinal mode in the linear and right-angle cavities, respectively. As a result, an orthogonally and linearly polarized dual-frequency laser at 1064 nm was obtained, and the oscillating-mode spectrum was observed experimentally by the use of a confocal scanning Fabry–Perot interferometer, as shown in Figure 29.

The tilt angle of the BC_2_ was maintained at a constant 2.5°, and the frequency difference of the dual-frequency laser at 1064 nm was tuned by adjusting the tilt angles of the BC_1_. The experimental relationship between the frequency difference and the tilt angles of the BC_1_ was obtained, as shown in Figure 30. It can be seen that the frequency difference of the dual-frequency laser at 1064 nm was tuned in a range from 5.2 GHz to 147.3 GHz when the BC_1_ was tilted from approximately 1.5° to 2.5°, and the maximum frequency difference reached nearly to the fluorescence linewidth of the Nd:YAG crystal.

In 2016, our group also reported a TCDFL with electro-optical modulators [37], as shown in Figure 31, which consisted of two standing-wave cavities that shared the same gain medium of Nd:YAG. An electro-optic birefringent filter consisting of PBS and lithium niobate (LN) was used not only to select longitudinal mode but also to tune frequency and frequency difference. As a result, the simultaneous operation of orthogonally and linearly polarized dual-frequency laser was obtained, as shown in Figure 32. The experimentally obtained results indicated that the frequency difference was tuned from 0 to 132 GHz by changing the DC voltages applied to LNs. When the pump power was 900 mW, the output powers from the linear and right-angle cavities were equal to 20 mW and 26 mW, respectively.

#### 2.3.2. TCDFL with a Twisted-Mode Configuration

In order to produce the dual-frequency laser with a tunable frequency difference at 1064 nm, in 2015, our group designed and investigated a diode-pumped TCDFL using a twisted-mode configuration [38], as shown in Figure 33, the two standing-wave cavities of which shared the same gain medium Nd:YAG, and the twisted-mode configuration reduced or even eliminated the spatial hole-burning effect of the gain so that the single-longitudinal-mode was oscillated in both standing-wave cavities. Thus, the orthogonally and linearly polarized dual-frequency laser at 1064 nm was obtained, and the oscillating mode spectrum was observed experimentally by the use of a confocal scanning F-P interferometer, as shown in Figure 34.

The experimental results showed that both cavities of the Nd:YAG laser oscillated steadily in single-longitudinal mode, and the frequency difference was tuned in a range from 0.3 GHz to 3 GHz by changing the cavity length. Theoretically, the maximum frequency difference was equal to one-half of the sum of both cavities’ longitudinal-mode intervals.

#### 2.3.3. TCDFL Using Intracavity F-P Etalon

The TCDFLs based on the principle of single-longitudinal-mode selection by the birefringent filter are not only simple but also have a low inserted cost. However, the spectral linewidths of the dual-frequency laser are not narrow enough due to the fact that the bandwidth of transmission peak of the birefringent filter is relatively wide, affecting the coherence lengths of the TCDFLs.

In 2022, our group reported a diode-pumped dual-frequency Nd:YAG laser with two standing-wave cavities sharing the common gain medium [40], which was based on the principle of single-longitudinal-mode selection by intracavity F-P etalon, as shown in Figure 35. With each of the cavities containing a piece of F-P etalon, the p-polarized and s-polarized components of the laser at 1064 nm were forced to oscillate simultaneously in single-longitudinal mode in both cavities, respectively. As a result, the orthogonally and linearly polarized dual-frequency laser at 1064 nm was output. Meanwhile, a coaxially propagating beam of the dual-frequency laser at 1064 nm escaped from the side of the intracavity PBS element because the PBS element had somewhat residual reflectance of the p-polarized beam and residual transmittance of the s-polarized beam.

The experimental results indicated that when both fused quartz F-P etalons with thicknesses of 0.5 mm and surface reflectivities of 90% were obliquely inserted into both cavities, respectively, the orthogonally and linearly polarized dual-frequency laser could be obtained by finely adjusting the tilt angles of the intracavity F-P etalons. The frequency difference tuning of the dual-frequency laser was realized by continually adjusting the tilt angles of both F-P etalons, and the typical oscillating spectrums of the dual-frequency laser were observed, as shown in Figure 36. It can be seen that the frequency difference was discontinuously tuned in a range from 16 to 76 GHz by finely adjusting the tilt angles of the intracavity F-P etalons, and theoretically, the maximum frequency difference was up to the oscillating bandwidth of the Nd:YAG laser. For the diode-pumped TCDFL with a frequency difference of 24 GHz, the threshold pump powers of the linear and right-angle cavities were equal to 1.9 W and 2 W, respectively, and the output powers of the linear and right-angle cavities were up to 229 and 190 mW, respectively.

## 3. Frequency-Difference-Stabilizing Systems for Dual-Frequency Solid-State Lasers

It is known that the accuracy of a synthetic-wave absolute-distance interferometric measurement ultimately depends on the maximum frequency difference of the dual-frequency laser and its stability. Therefore, the research on the frequency-difference-stabilizing technology of dual-frequency solid-state lasers is of great significance.

### 3.1. Frequency Difference Stabilization of Birefringent Dual-Frequency Solid-State Lasers

As described above, a kind of birefringent dual-frequency solid-state lasers has a single-axis configuration, and the frequency difference or beat-note stabilization has been investigated [46,47,48,49,50], the methods of which mainly include frequency-shifted optical feedback, saturable absorption, etc.

In 2007, the research group of L. Kervevan of France reported an original approach to stabilize the beat-note of a 1.53 μm dual-frequency Yb:Er glass laser via an optical self-injection process [51], as shown in Figure 37, in which the dual-frequency Yb:Er glass laser could output a dual-frequency laser at 1.53 μm with a beat-note of nearly 170 MHz. The optical self-injection process consisted of selecting one of the two linear modes as a master oscillator with a polarization filter, then frequency shifting the optical wave using an external acousto-optic modulator, and finally, using it to inject the other mode.

When the frequency-shifted optical beam was correctly reinjected into the oscillating mode of the laser cavity, the stability of the locking technique was tested by recording the beat note versus the synthesizer frequency, as shown in Figure 38a. It corresponded to a slow frequency deviation of about 0.27 mHz/s. A linear fit was applied to the measured synthesizer frequency [straight line in Figure 38a], and the difference between this reference line and the optical beat-note was observed in the histogram plotted curve in Figure 38b. It can be seen that a fitted Gaussian distribution curve allowed for estimating the stability of the frequency locking to be less than 0.25 Hz.

### 3.2. Frequency Difference Stabilization of TCDFL

Due to the fact that the TCDFL has both standing-wave cavities with two separate output couplers, it is easy to actively stabilize the resonant frequency of each cavity to a common frequency reference so that a high stability of the frequency difference can be obtained.

As a commonly used method, the PDH frequency-stabilization method integrates with the technologies of both electro-optic phase modulation and optical heterodyne detection. In the past few decades, the laser-frequency-stabilizing technologies based on the PDH method have been widely investigated worldwide due to their advantages of fast servo response, low noise, and high-frequency stability [52,53,54,55,56,57].

#### 3.2.1. Double-Modulator QD-PDH Frequency-Difference Stabilizing System for TCDFL

In 2022, our group reported a frequency-difference-stabilizing system for the diode-pumped TCDFL at 1064 nm using a double-modulator QD-PDH frequency-stabilizing method [43], as shown in Figure 39, which included two sets of QD-PDH frequency-stabilizing subsystems (see parts II and III) that shared the same F-P cavity as the frequency reference, and the magnitude of the frequency difference was required to be an integer number representing times of the free spectral range (FSR) of the referenced F-P cavity.

A QD-PDH method-based frequency-difference-stabilizing system for the diode-pumped TCDFL with a frequency difference of 24 GHz at 1064 nm was established and investigated, in which the free spectral range and the finesse of the referenced F-P cavity were equal to 375 MHz and 421, respectively. Both frequencies of the TCDFL were successfully frequency stabilized to the two different resonant frequencies of the F-P cavity during a period of about 1 h, the error signals of the two QD-PDH frequency-stabilizing subsystems were obtained experimentally, as shown in Figure 40, the maximum offset voltages of the QD-PDH error signals were equal to 77 mV and 74.2 mV, respectively, and correspondingly, the laser-frequency drifts of the linear and right-angle cavities were determined to be <0.35 MHz and 0.36 MHz, respectively. The frequency-difference fluctuations of the frequency-locked TCDFL are shown in Figure 41, and the maximum change in the frequency difference was <0.55 MHz. According to the Allan variance, the laser-frequency stabilities of the linear and right-angle cavities were better than 2.3 × 10^−11^ and 2.7 × 10^−11^, respectively, corresponding to a frequency-difference stability better than 4.2 × 10^−7^.

#### 3.2.2. Single-Modulator QD-PDH Frequency-Difference-Stabilizing System for TCDFL

In 2022, our group proposed a new scheme of the phase modulation of the orthogonally and linearly polarized dual-frequency laser using a single electro-optic modulator (EOM), and a simple frequency-difference-stabilizing system for the TCDFL using a single-modulator QD-PDH frequency-stabilizing method was designed [44], as shown in Figure 42, which included two sets of QD-PDH frequency-stabilizing subsystems (see parts II and III) that shared the same electro-optic phase modulation unit and the same frequency reference of the F-P cavity.

A QD-PDH frequency-difference-stabilizing system for the same diode-pumped TCDFL with a frequency difference of 24 GHz at 1064 nm was established and investigated. Both frequencies of the TCDFL at 1064 nm were successfully frequency-stabilized to the two different resonant frequencies of the referenced F-P cavity during a period of about 1 h, the error signals of the two QD-PDH frequency-stabilizing subsystems were obtained experimentally, as shown in Figure 43, the maximum offset voltages of the QD-PDH error signals were equal to 84.5 mV and 76.7 mV, respectively, and correspondingly, the laser-frequency drifts of the linear and right-angle cavities were determined to be <0.34 and 0.35 MHz, respectively. The frequency-difference fluctuations of the frequency-locked TCDFL are shown in Figure 44, and the maximum change in the frequency difference was <0.51 MHz. According to the Allan variance, the laser-frequency stabilities of the linear and right-angle cavities were better than 1.6 × 10^−11^ and 2.0 × 10^−11^, respectively, corresponding to a frequency-difference stability better than 2.9 × 10^−7^.

The experimental results obtained above indicate that compared with the double-modulator QD-PDH frequency-difference-stabilizing system shown in Figure 39, the single-modulator QD-PDH frequency-difference-stabilizing system shown in Figure 42 is not only simple, but also has better performances in the linear-dynamic range, frequency-discriminating sensitivity, frequency stabilization, and frequency-difference stabilization, as listed in Table 1.

## 4. Summary and Developing Trends

In order to achieve the high-accuracy synthetic-wave absolute-distance measurement, a frequency-difference-stabilized dual-frequency solid-state laser with a tunable and large frequency difference is needed to use as a light source to form a number of stable multistage synthetic wavelengths. We reviewed the advances in research on different kinds of dual-frequency solid-state lasers, including birefringent dual-frequency solid-state lasers, biaxial dual-frequency Yb:KGW lasers, and TCDFLs based on the different principles of single-longitudinal-mode selection. Additionally, several typical frequency-difference-stabilizing systems for dual-frequency solid-state lasers have been introduced and analyzed.

Taking into account of the requirements of synthetic-wave absolute-distance interferometry and the present research status of dual frequency, we may predict future development trends in dual-frequency solid-state lasers, mainly including the following aspects:

First of all, the new oscillating principles and methods of orthogonally and linearly polarized dual-frequency solid-state lasers will gain attraction for investigation. Especially, the different principles and methods of single-longitudinal-mode selection should be taken into account. In addition to the standing-wave TCDFLs described above, traveling-wave TCDFLs can also be designed and investigated.

Secondly, the new technologies for enlarging the frequency-difference-tuning range of dual-frequency solid-state lasers will be researched and developed. Intracavity second-harmonic generation (SHG) can be used to enlarge the frequency-difference-tuning range, due to the fact that the frequency difference of second-harmonic dual-frequency lasers is theoretically twice that of fundamental dual-frequency lasers.

Thirdly, the new technologies of the spectral-line narrowing of dual-frequency solid-state lasers will be researched and developed. The spectral linewidths of dual-frequency solid-state lasers determine the coherence lengths, which affect the measuring range of the synthetic-wave absolute-distance interferometry.

Finally, the new technologies of frequency difference stabilization of dual-frequency solid-state lasers will be researched and developed. The scheme of frequency difference stabilization depends on the dual-frequency solid-state laser configuration, and for TCDFL, the frequency difference stability of the dual-frequency laser will be further improved, provided that some special measurements be taken, such as using a highly stable F-P reference cavity with high finesse, optimizing the performances of the feedback control system, etc. It is worth mentioning that the femtosecond optical-frequency comb has a wide linewidth and a high frequency stability, which can be used as the frequency reference of the frequency-difference-stabilizing system.

## Figures and Tables

**Figure 1 sensors-23-03206-f001:**
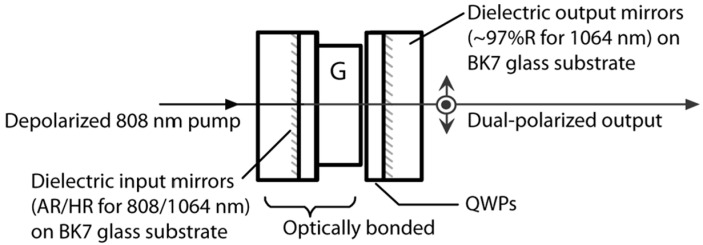
Schematic diagram of dual-frequency ceramic microchip Nd:YAG laser [20]. QWP: quarter-wave plate; G: glass.

**Figure 2 sensors-23-03206-f002:**
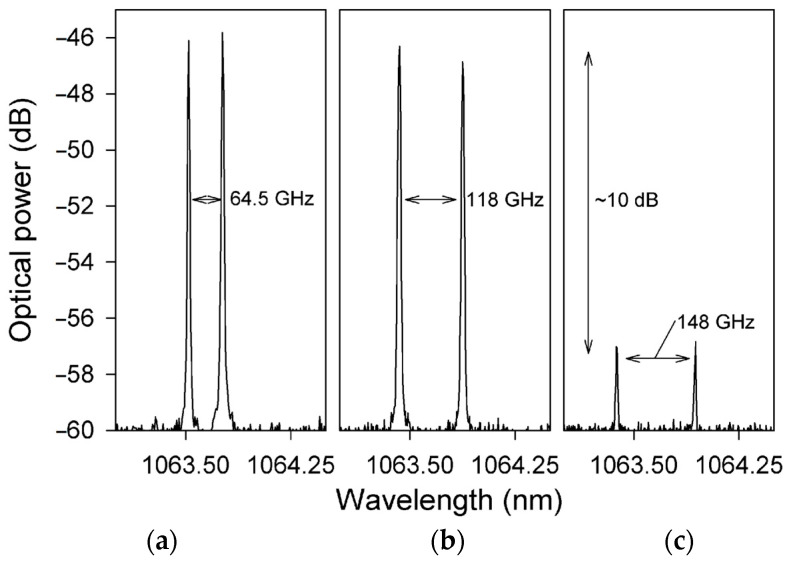
Oscillating spectrums of dual-frequency laser with difference frequency differences of (**a**) 64.5 GHz, (**b**) 118 GHz, and (**c**) 148 GHz [20].

**Figure 3 sensors-23-03206-f003:**
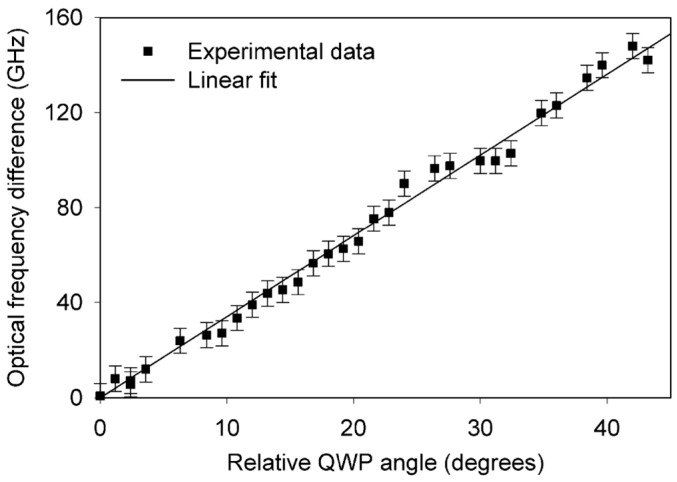
Tuning of the optical frequency separation between orthogonally polarized modes as a function of relative quarter-wave-plate angles [20].

**Figure 4 sensors-23-03206-f004:**
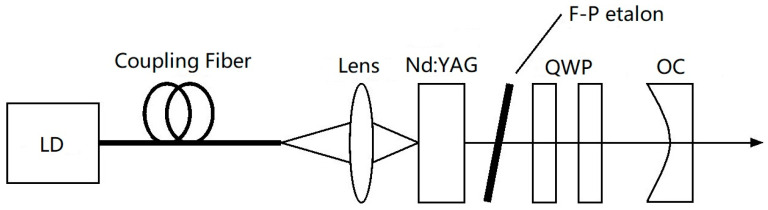
Schematic diagram of dual-frequency Nd:YAG laser [21]. LD: laser diode; F-P etalon: Fabry–Perot etalon; QWP: quarter-wave plate; OC: output coupler.

**Figure 5 sensors-23-03206-f005:**
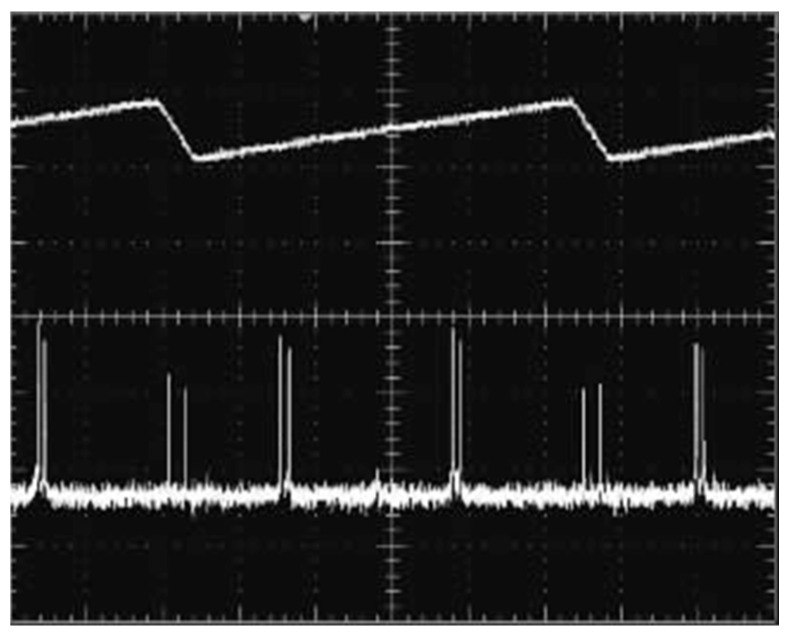
Oscillating spectrum of dual-frequency laser [21].

**Figure 6 sensors-23-03206-f006:**
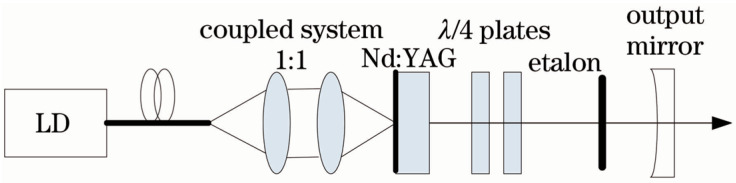
Diagram of the tunable two-frequency solid-state laser with coupled cavities [22]. LD: laser diode; λ/4 plate: quarter-wave plate.

**Figure 7 sensors-23-03206-f007:**
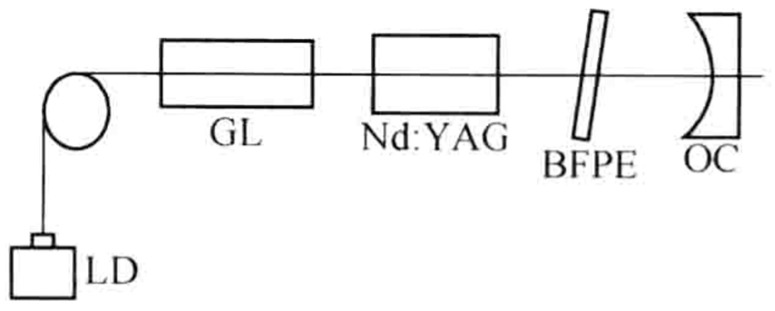
Experimental setup of diode-pumped birefringent dual-frequency Nd:YAG laser [23]. LD: laser diode; GL: gradient-index lens; BFFP: birefringent Fabry–Perot etalon; OC: output coupler.

**Figure 8 sensors-23-03206-f008:**
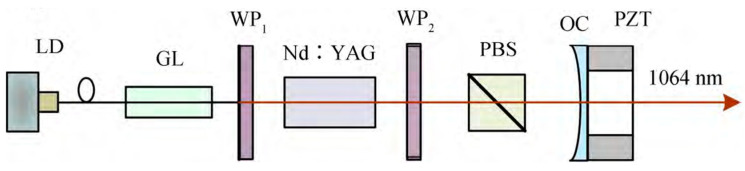
Schematic diagram of dual-frequency Nd:YAG laser with a detuning twisted-mode cavity [24]. LD: laser diode; GL: gradient-index lens; WP: wave plate; PBS: polarizing beam splitter; OC: output coupler; PZT: piezoelectric tube.

**Figure 9 sensors-23-03206-f009:**
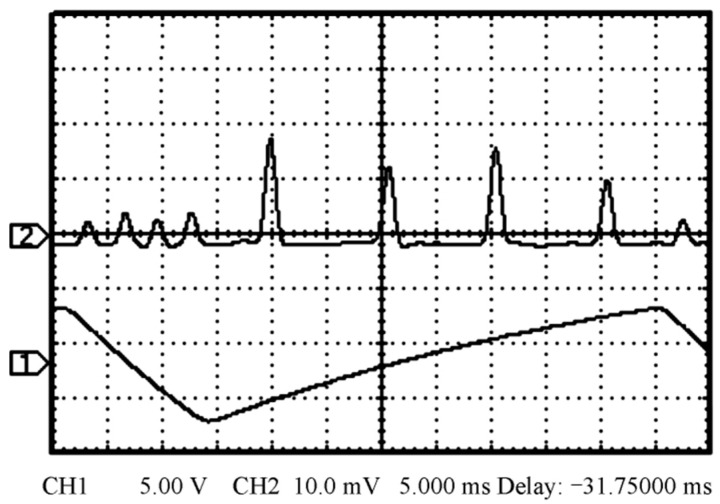
Oscillating spectrum of dual-frequency laser [24].

**Figure 10 sensors-23-03206-f010:**
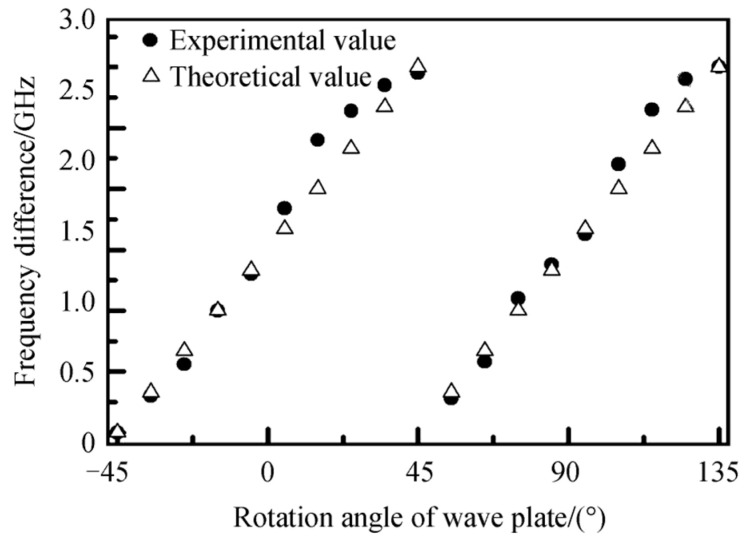
Dependence of mode-splitting magnitude on the rotation angle of quarter-wave plate [24].

**Figure 11 sensors-23-03206-f011:**
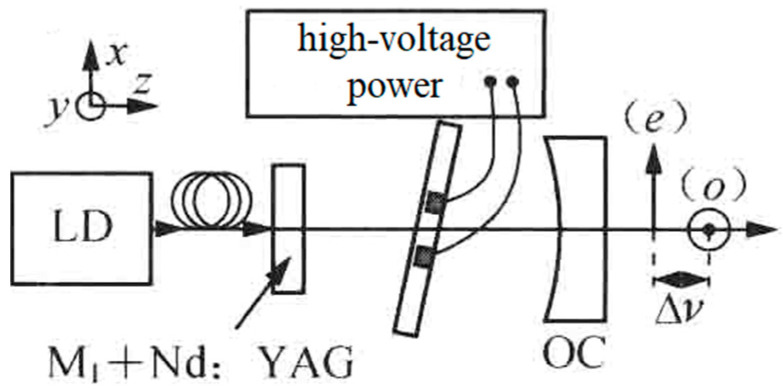
Schematic diagram of electro-optical birefringent dual-frequency Nd:YAG laser [25]. LD: laser diode; M: mirror; OC: output coupler.

**Figure 12 sensors-23-03206-f012:**
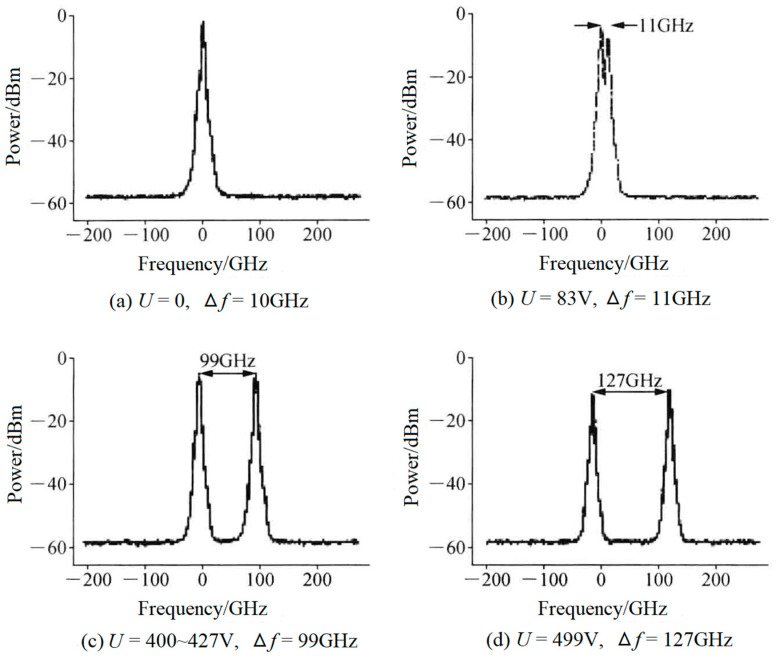
Oscillating spectrum of dual-frequency laser with different frequency difference [25].

**Figure 13 sensors-23-03206-f013:**
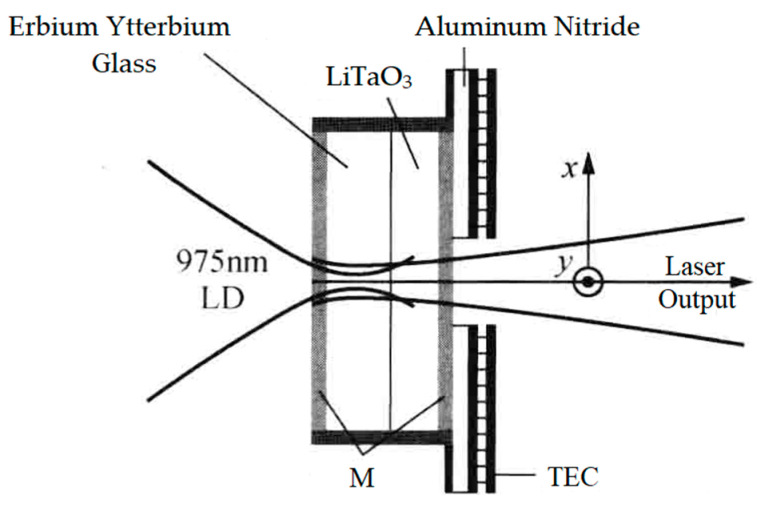
Schematic diagram of thermo-optical birefringent microchip dual-frequency laser [26]. LD: laser diode; M: mirror; TEC: thermo-electric cooler.

**Figure 14 sensors-23-03206-f014:**
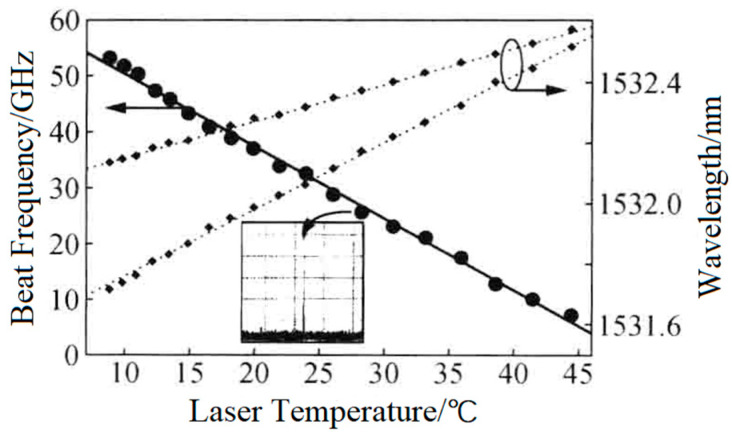
Dependence of oscillating wavelength and its beat frequency on laser temperature [26].

**Figure 15 sensors-23-03206-f015:**
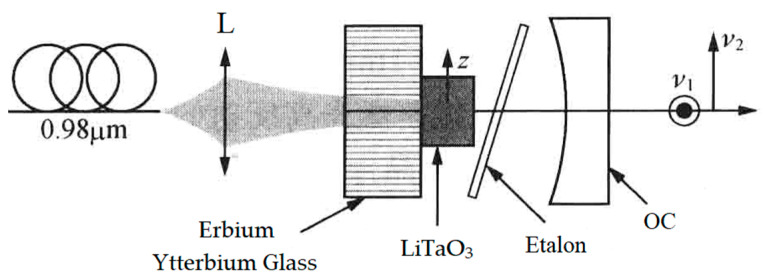
Schematic diagram of dual-frequency erbium–ytterbium glass laser [27]. L: lens; OC: output coupler.

**Figure 16 sensors-23-03206-f016:**
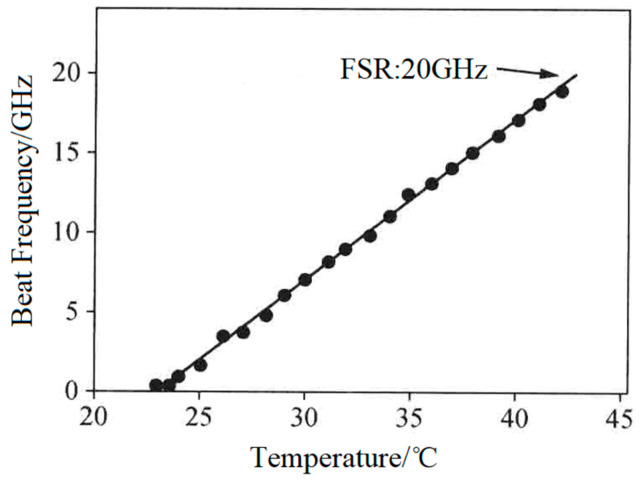
Dependence of frequency difference of dual-frequency laser on laser temperature [27].

**Figure 17 sensors-23-03206-f017:**
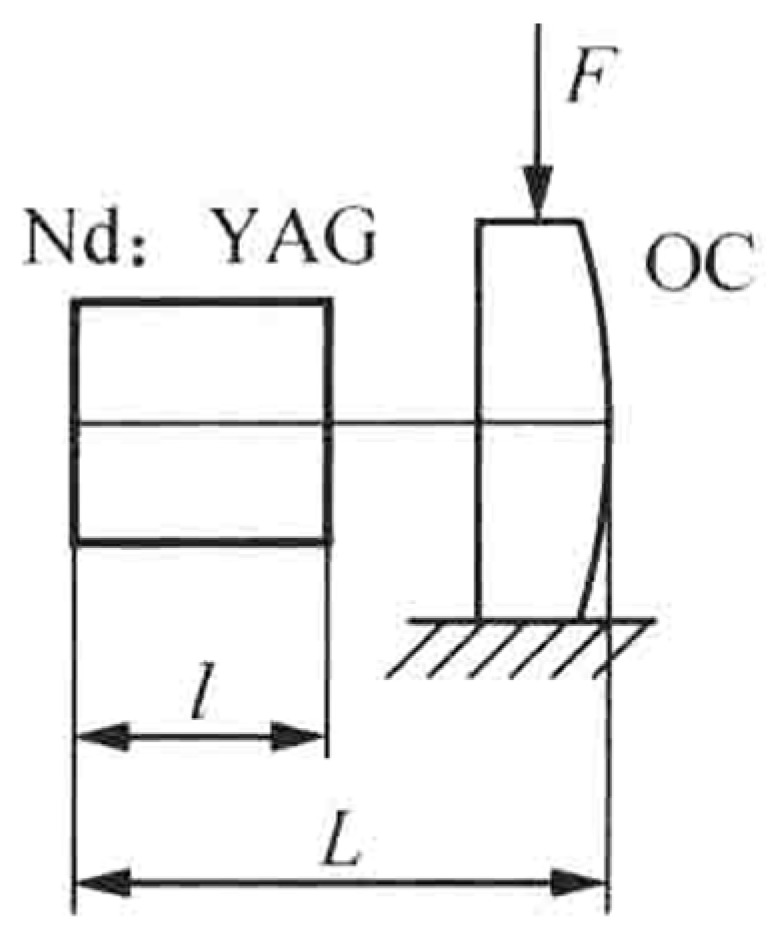
Schematic diagram of coupled-cavity dual-frequency Nd:YAG laser [28]. OC: output coupler.

**Figure 18 sensors-23-03206-f018:**
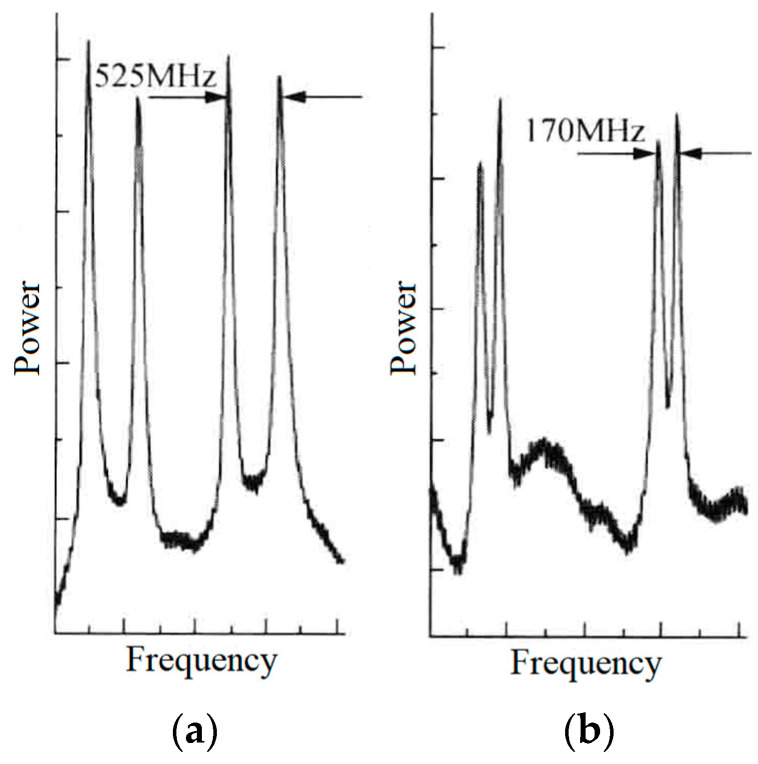
Oscillating spectrum of dual-frequency laser with different frequency difference of (**a**) 525 MHz and (**b**) 170 MHz [28].

**Figure 19 sensors-23-03206-f019:**
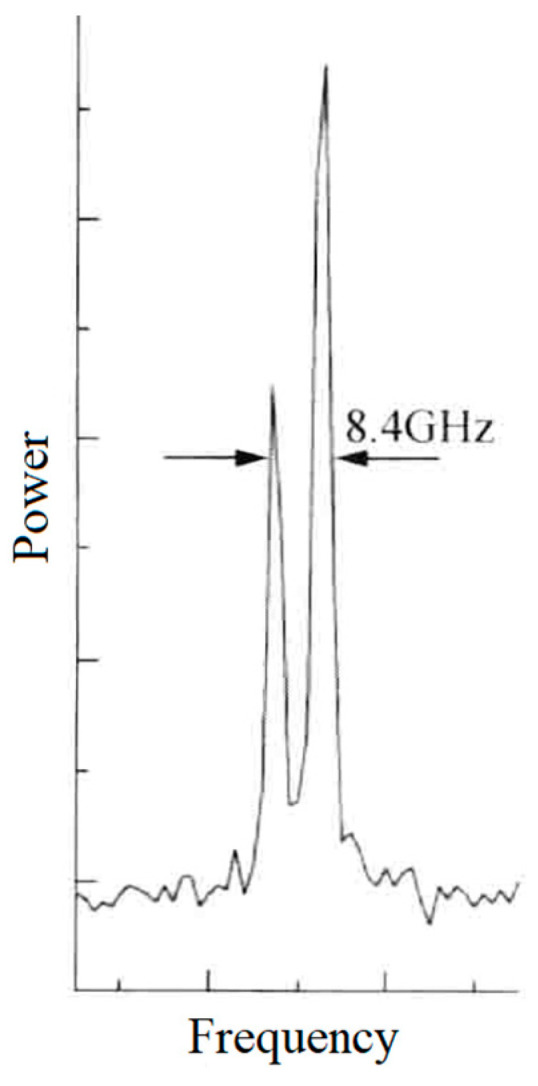
Oscillating spectrum of orthogonally polarized dual-frequency Nd:YAG laser [28].

**Figure 20 sensors-23-03206-f020:**
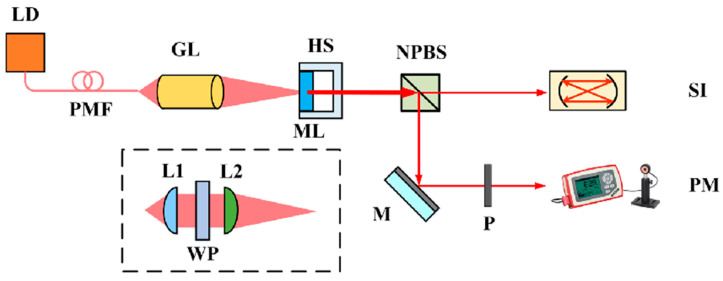
Schematic setup of the dual-frequency microchip laser [29]. LD: laser diode; PMF: polarization-maintaining fiber; GL: grin lens; L1, L2: lenses; WP: half waveplate; HS: heat sink; ML: Nd:YAG microchip laser; NPBS: nonpolarizing beam splitter; SI: scanning interferometer; PM: power meter; M: reflective mirror; P: polarizer.

**Figure 21 sensors-23-03206-f021:**
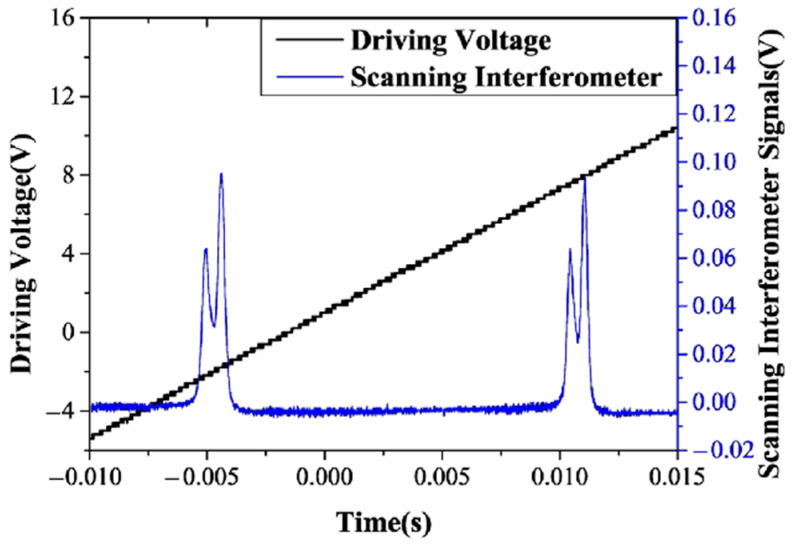
Oscillating spectrum of dual-frequency laser (blue trace) and triangular-wave voltage (black trace) [29].

**Figure 22 sensors-23-03206-f022:**
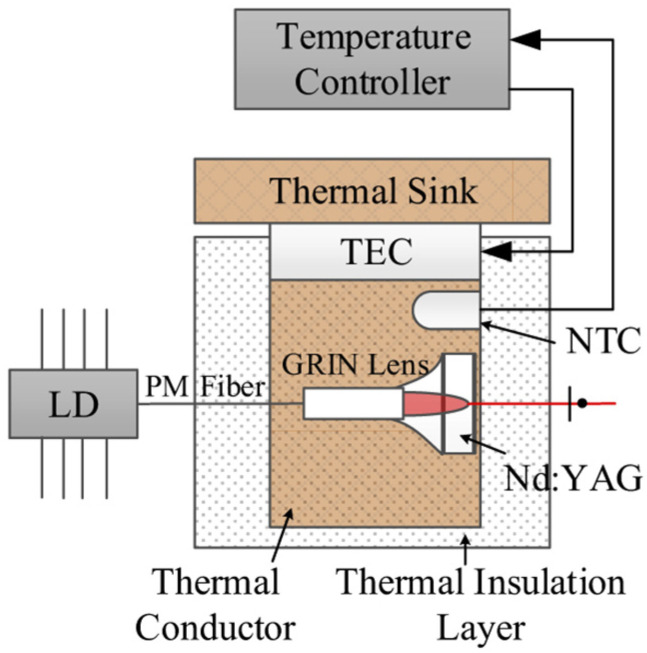
Schematic of microchip Nd:YAG dual-frequency laser [30]. LD: laser diode; TEC: thermo-electric cooler; NTC: negative-temperature-coefficient thermistor.

**Figure 23 sensors-23-03206-f023:**
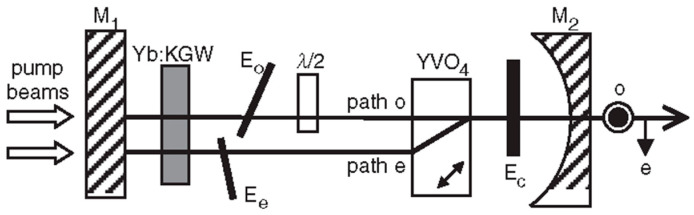
Schematic diagram of dual-frequency Yb:KGW laser [32]. E_o_, E_e_, E_c_: etalon; M_1_: plane-dichroic mirror; M_2_: output coupler.

**Figure 24 sensors-23-03206-f024:**
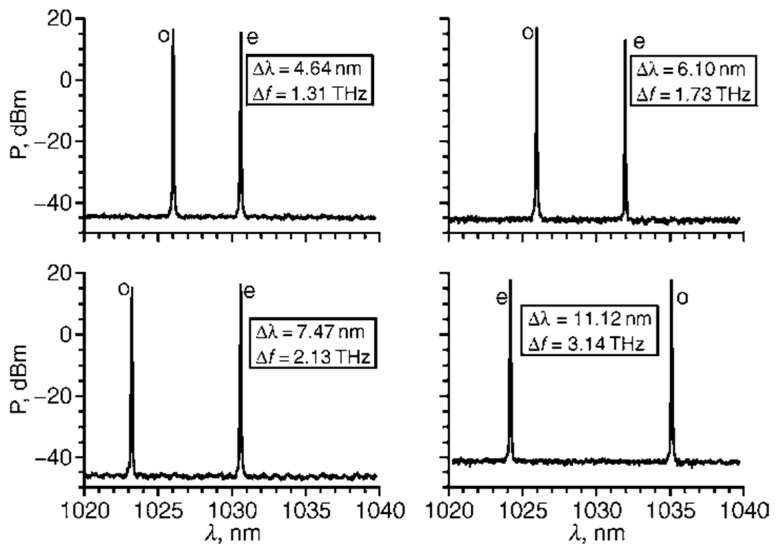
Measured optical spectra of dual-frequency laser when tuned for various frequency differences [32].

**Figure 25 sensors-23-03206-f025:**
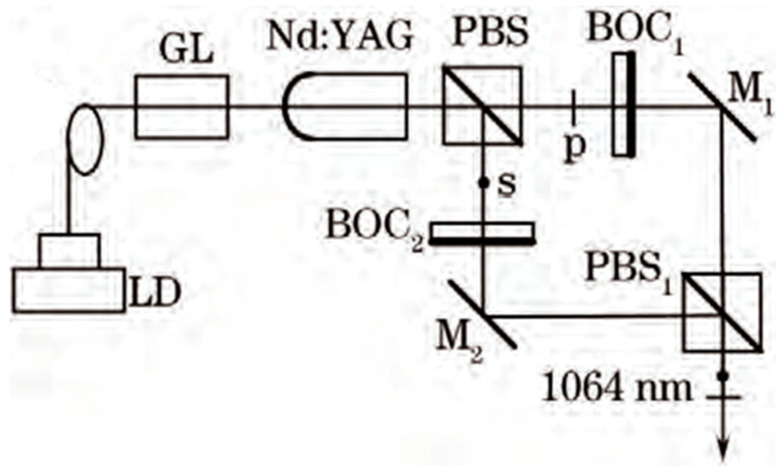
Schematic diagram of dual-frequency Nd:YAG laser system at 1064 nm [35]. LD: laser diode; GL: gradient-index lens; PBS: polarizing beam splitter; BOC: birefringent output coupler; M: mirror.

**Figure 26 sensors-23-03206-f026:**
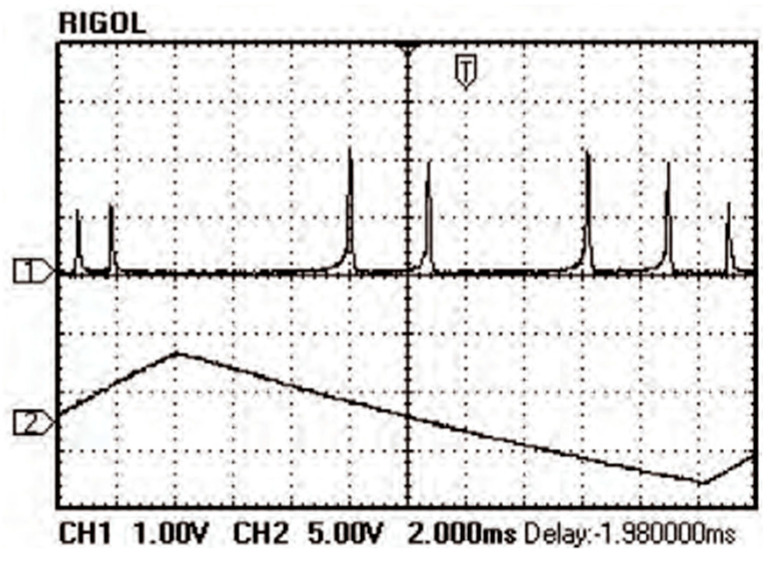
Mode patterns of simultaneous dual-frequency oscillation [35].

**Figure 27 sensors-23-03206-f027:**
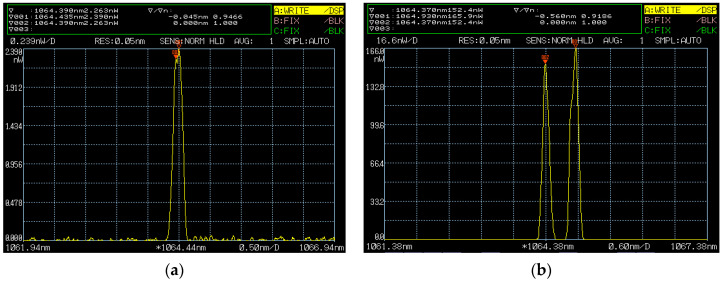
Oscillating spectra of dual-frequency laser with different frequency differences of (**a**) 11.9 GHz and (**b**) 148.4 GHz [35].

**Figure 28 sensors-23-03206-f028:**
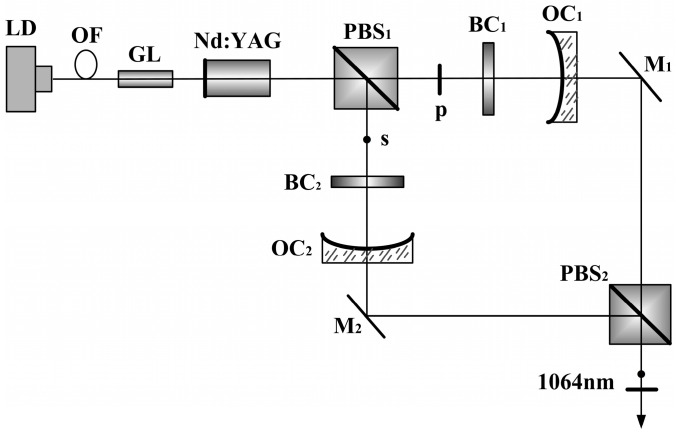
Schematic diagram of TCDFL based on the principle of longitudinal-mode selection by birefringent filter [36]. LD: laser diode; OF: optical fiber; GL: gradient-index lens; PBS: polarizing bean splitter; BC: birefringent crystal; OC: output coupler; M: mirror.

**Figure 29 sensors-23-03206-f029:**
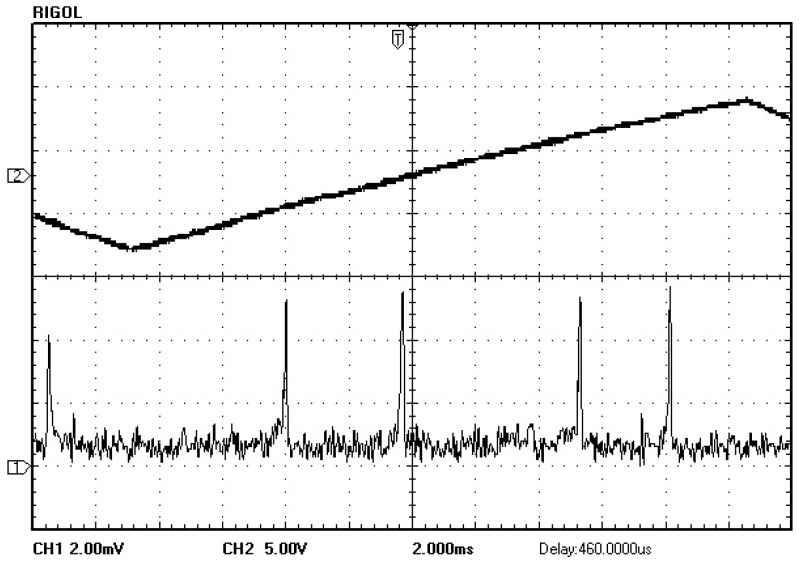
Oscillating mode spectrum of dual-frequency laser [36].

**Figure 30 sensors-23-03206-f030:**
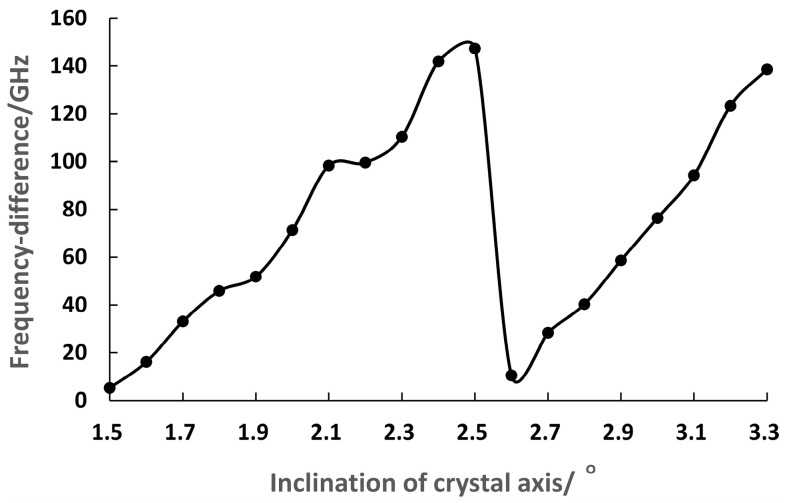
Relationship between frequency difference and tilt angle of the BC_1_ [36].

**Figure 31 sensors-23-03206-f031:**
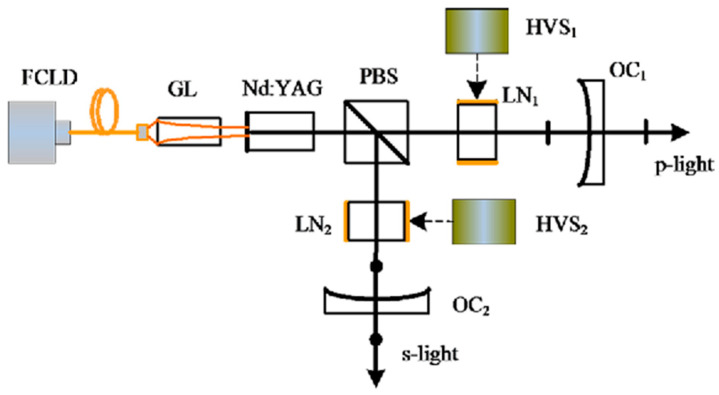
Schematic of two-cavity dual-frequency Nd:YAG laser [37]. FCLD: fiber-coupled laser diode; GL: gradient-index lens; PBS: polarizing bean splitter; LN: lithium niobate; HVS: high-voltage source; OC: output coupler.

**Figure 32 sensors-23-03206-f032:**
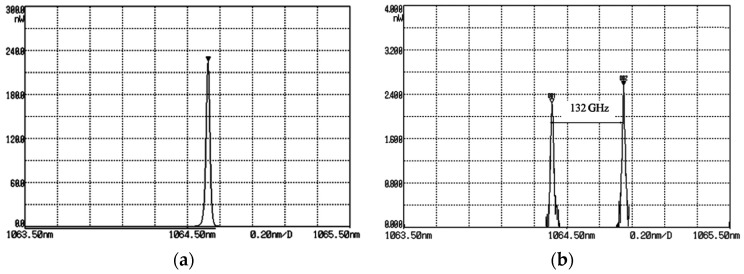
Oscillating spectrum of dual-frequency laser with different frequency differences of (**a**) 0 GHz and (**b**) 132 GHz [37].

**Figure 33 sensors-23-03206-f033:**
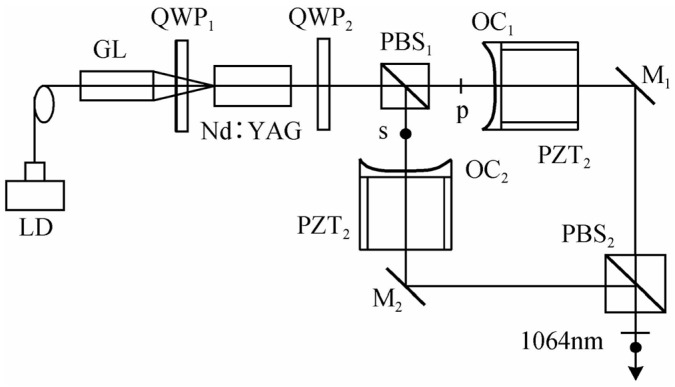
Schematic diagram of diode-pumped TCDFL using the twisted-mode configuration [38]. LD: laser diode; GL: gradient-index lens; QWP: quarter-wave plate; PBS: polarizing beam splitter; OC: output coupler; PZT: piezoelectric transducer; M: mirror.

**Figure 34 sensors-23-03206-f034:**
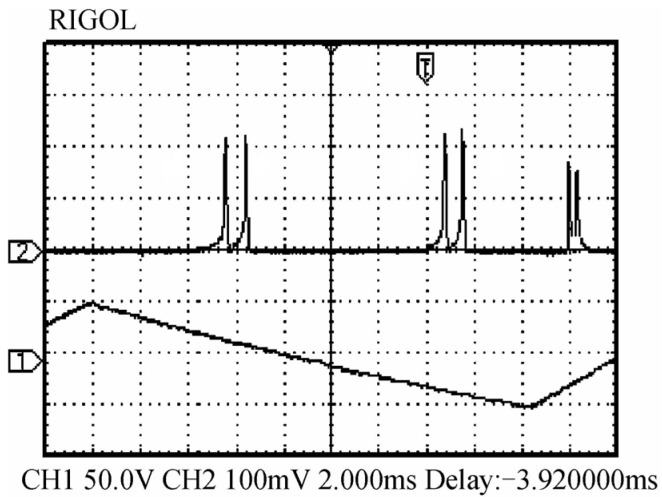
Oscillating-mode spectrum of dual-frequency laser [38].

**Figure 35 sensors-23-03206-f035:**
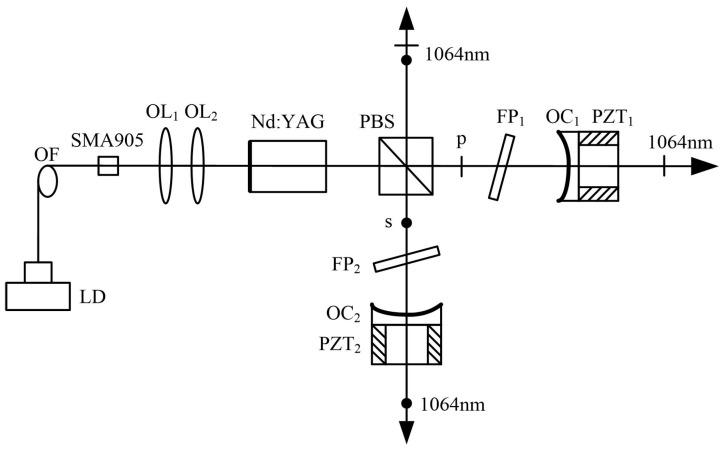
Schematic of TCDFL based on the principle of longitudinal-mode selection by single F-P etalon [40]. LD: laser diode; OF: optical fiber; SMA905: OF connector; OL: optical lens; PBS: polarizing beam splitter; FP: F-P etalon; OC: output coupler; PZT: piezoelectric transducer.

**Figure 36 sensors-23-03206-f036:**
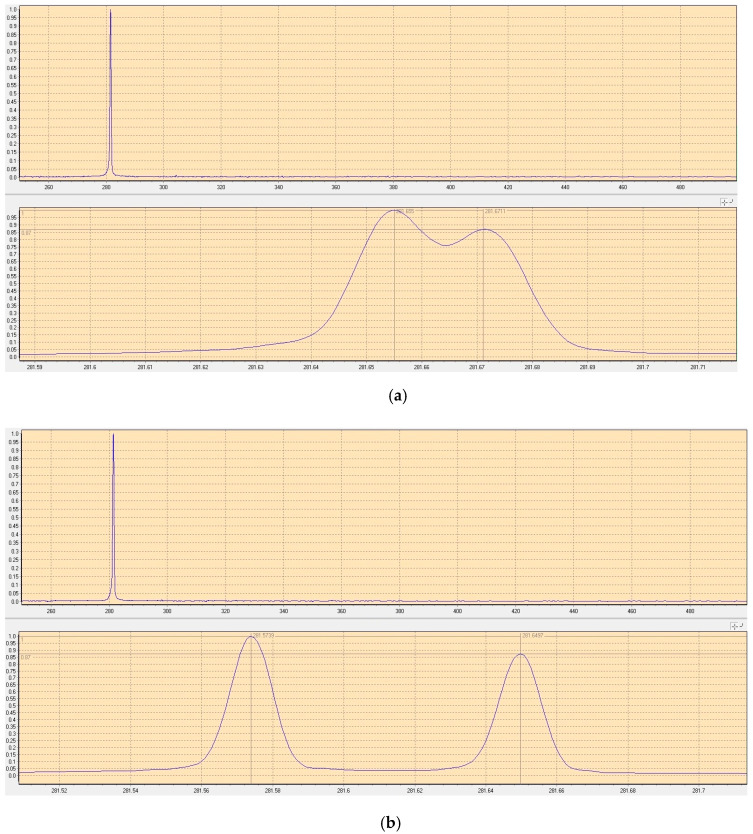
Oscillating spectrum of dual-frequency laser with different frequency differences of (**a**) 16 GHz and (**b**) 76 GHz [40].

**Figure 37 sensors-23-03206-f037:**
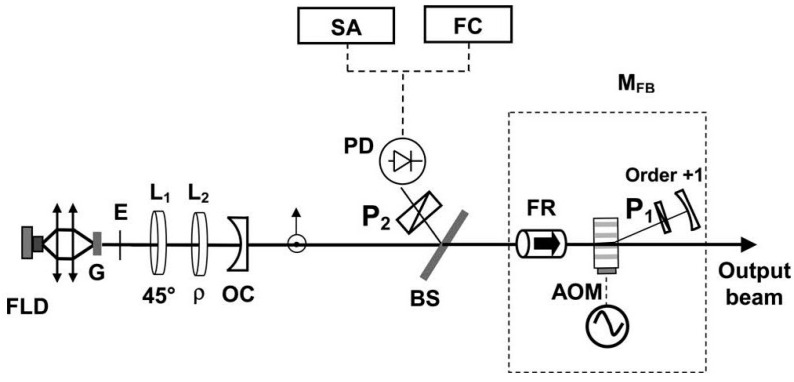
Schematic of self-injection stabilization process for 1.53 μm dual-frequency phosphate glass laser [51]. FLD: fiber-pigtailed laser diode; G: Yb:Er glass plate; E: intracavity etalon; L_1_ and L_2_: quarter-wave plates; P_1_ and P_2_: linear polarizers; OC: output coupler; BS: beam splitter; FR: Faraday rotator; AOM: acousto-optic modulator; PD: photodiode; SA: spectrum analyzer; FC: frequency counters; M_FB_: optical feedback module; ρ: angular adjustment.

**Figure 38 sensors-23-03206-f038:**
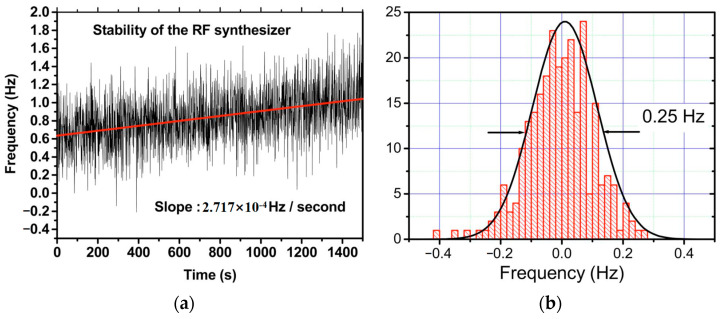
Schematic of self-injection stabilization process for 1.53 μm dual-frequency phosphate glass laser [51]. (**a**) Instantaneous fluctuations and long-term deviation of the synthesizer frequency versus time. (**b**) Histogram of the temporal stability of the beat note between the two orthogonal modes using the locking technique based on the frequency-shifted optical feedback loop.

**Figure 39 sensors-23-03206-f039:**
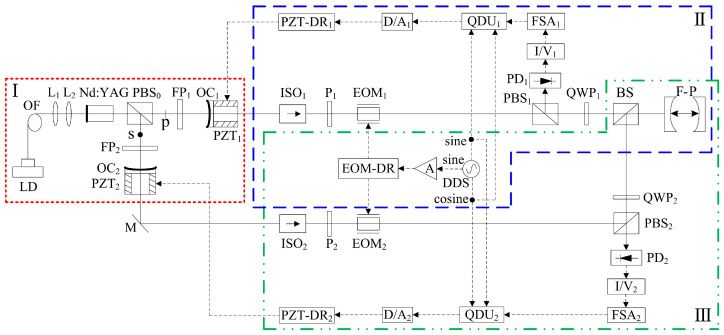
Schematic diagram of the frequency-difference-stabilizing system for the TCDFL using the double-modulator QD-PDH frequency-stabilizing method [43]. LD: laser diode; OF: optical fiber; L: lens; PBS: polarizing beam splitter; FP: F-P etalon; OC: output coupler; PZT: piezoelectric transducer; M: mirror; ISO: optical isolator; P: polarizer; EOM: electro-optic modulator; DDS: direct digital synthesizer; A: amplifier; EOM-DR: EOM driver; QWP: quarter-wave plate; BS: beam splitter; F-P: F-P reference cavity; PD: photodetector; I/V: current-to-voltage; FSA: frequency-selective amplifier; QDU: quadrature-demodulated unit; D/A: digital-to-analog; PZT-DR: PZT driver.

**Figure 40 sensors-23-03206-f040:**
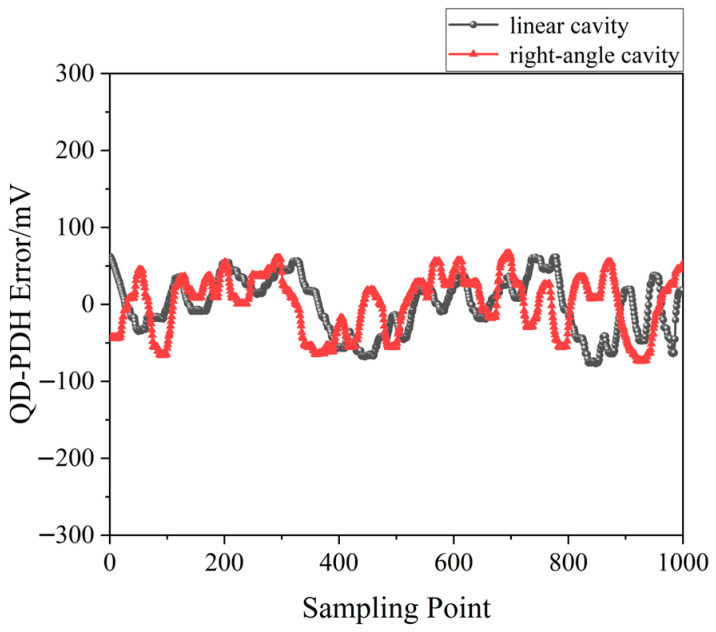
Error signals of the frequency-locked QD-PDH subsystems [43].

**Figure 41 sensors-23-03206-f041:**
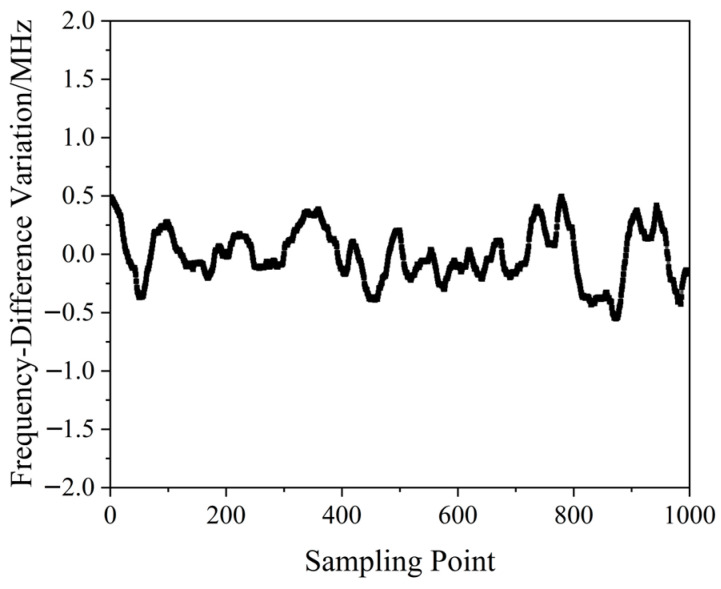
Frequency difference variation in the frequency-locked dual-frequency Nd:YAG laser [43].

**Figure 42 sensors-23-03206-f042:**
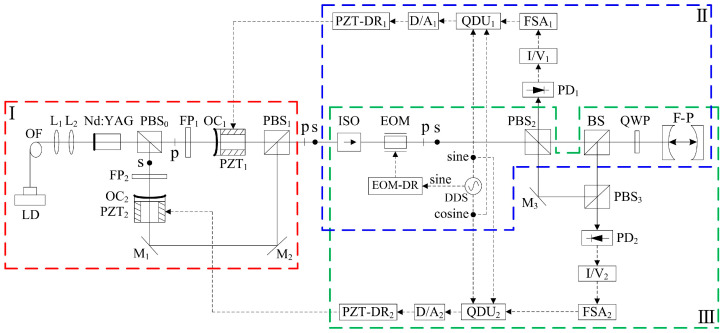
Schematic diagram of the frequency-difference-stabilizing system for the TCDFL using the single-modulator QD-PDH frequency-stabilizing method [44]. LD: laser diode; OF: optical fiber; L: lens; PBS: polarizing beam splitter; FP: F-P etalon; OC: output coupler; PZT: piezoelectric transducer; M: mirror; ISO: optical isolator; EOM: electro-optic modulator; DDS: direct digital synthesizer; EOM-DR: EOM driver; BS: beam splitter; QWP: quarter-wave plate; F-P: F-P reference cavity; PD: photodetector; I/V: current-to-voltage; FSA: frequency-selective amplifier; QDU: quadrature-demodulated unit; D/A: digital-to-analog; PZT-DR: PZT driver.

**Figure 43 sensors-23-03206-f043:**
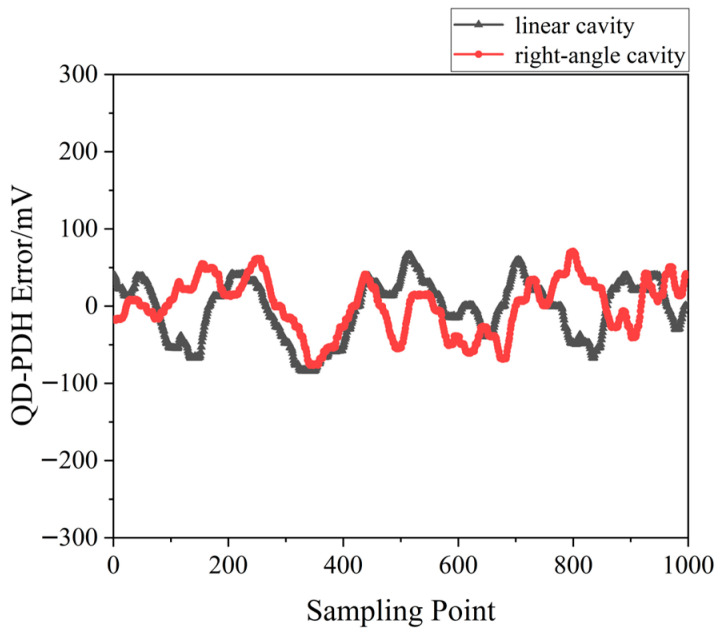
Error signals of the frequency-locked QD-PDH subsystems [44].

**Figure 44 sensors-23-03206-f044:**
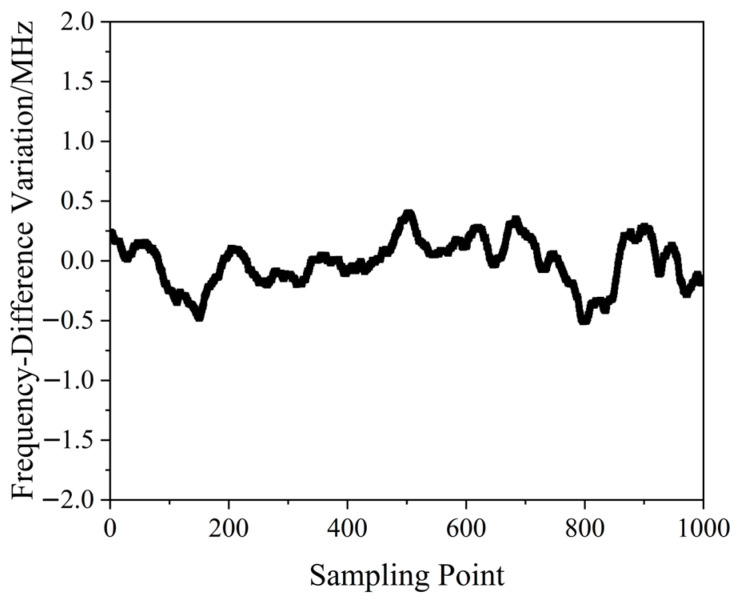
Frequency difference variation in the frequency-locked dual-frequency Nd:YAG laser [44].

**Table 1 sensors-23-03206-t001:** Comparison of experimental results of two QD-PDH frequency-difference-stabilizing systems [44].

Frequency-Stabilizing System	Resonant Cavity	Linear Dynamic Range/MHz	Frequency- Discriminating Sensitivity mV/MHz	Frequency Stability	Frequency- Difference Stability
Single-modulator QD-PDH	Linear cavity	5.36	251.71	1.6 × 10^−11^	2.9 × 10^−7^
	Right-angle cavity	5.25	222.34	2.0 × 10^−11^	
Double-modulator QD-PDH	Linear cavity	5.08	222.72	2.3 × 10^−11^	4.2 × 10^−7^
	Right-angle cavity	4.87	208.98	2.7 × 10^−11^	

## Data Availability

Data are available on request due to restrictions, e.g., privacy or ethical.

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
