# Peer review of "Advances of Research on Dual-Frequency Solid-State Lasers for Synthetic-Wave Absolute-Distance Interferometry"

_sensors, 2023, doi:10.3390/s23063206_

Round 1
Reviewer 1 Report
This research describes two-cavity dual-frequency Nd:YAG lasers. The major contents are frequency stabilization and absolute-distance interferometer application. However, there are some problems as follows, please comment.
1. There are more than 12 references are published by the author of this manuscript. Especially, for the recent research about the two-cavity dual-frequency Nd:YAG lasers, the proportion of the self-citation is unreasonable. It is not asking the author to remove the self-citation reference. On the contrary, I recommend adding more references that are published by the different research teams. The references are from the same research group, so there will be too much homogeneity in this article. For a review paper, the diversity of viewpoints in every research team or reference is very important.
2. For a review paper, the topic of this manuscript is interesting, but the scope of the content seems insufficient. This comment also refers to the “Instructions for Authors” of Sensors. According to the instruction of the review paper, this manuscript seems not to fit the requirement. The major conclusion could be obtained from a few of the references, e.g., references [30, 31, 32]. Please extend the scope of this manuscript, e.g., the frequency stabilization methods, the bottlenecks of various technologies, or the discussion of the different application fields.
3. Some self-citation is duplicated, e.g., [26-27] and [28-29]. These references are cited in Lines 286 and 291, and the technologies of these references seem similar. If it is not necessary, please cite the latest or the most significant reference. Otherwise, it should have some description of the difference between those references.
Reviewer 2 Report
This is a nice review article. Congrats.
Lines 241-243: please revisit, the English doesn't read well.
As a result, ..... output
Line 311: what are the units after 421?
Line 400: a candidate (not an candidate)
Reviewer 3 Report
1. Authors should summarize research progress more objectively and reduce statements such as: our group.
2. Citations to the reported works should be more explicit and annotated, especially for the graphs
3.There remains some spelling and grammar mistakes, that the author needs to polish the language.
Author Response
You replied to the comments.

Round 2
Reviewer 1 Report
After reviewing the revised manuscript, I am pleased to say that the quality and diversity of this paper have significantly improved. As a result, I recommend it for publication.